# Quantitative assessment of the universal thermopower in the Hubbard model

Wen O. Wang [1,2] ✉, Jixun K. Ding [1,2], Edwin W. Huang [3,4,5], Brian Moritz [2] & Thomas P. Devereaux [2,6,7] ✉

As primarily an electronic observable, the room-temperature thermopower $S$ in cuprates provides possibilities for a quantitative assessment of the Hubbard model. Using determinant quantum Monte Carlo, we demonstrate agreement between Hubbard model calculations and experimentally measured room-temperature $S$ across multiple cuprate families, both qualitatively in terms of the doping dependence and quantitatively in terms of magnitude. We observe an upturn in $S$ with decreasing temperatures, which possesses a slope comparable to that observed experimentally in cuprates. From our calculations, the doping at which $S$ changes sign occurs in close proximity to a vanishing temperature dependence of the chemical potential at fixed density. Our results emphasize the importance of interaction effects in the systematic assessment of the thermopower $S$ in cuprates.

The Hubbard model, despite decades worth of study, remains enigmatic as a model to describe strongly correlated systems. Due to the fermion sign problem and exponential complexity, only one-dimensional systems have lent themselves to error-free estimations of ground states and their properties. Recently, angle-resolved photo-emission studies have demonstrated that a one-dimensional Hubbard-extended Holstein model can quantitatively reproduce spectra near the Fermi energy[1–3]. In two dimensions, the community lacks exact results in the thermodynamic limit; nevertheless, many of the extracted properties from simulations of the Hubbard model bear a close resemblance to observables measured in experiments, particularly those performed on high-temperature superconducting cuprates. These properties include the appearance of antiferromagnetism near half-filling, stripes, and strange metal behavior[4–6]. However, quantitative assessments have remained out of reach, particularly regarding transport properties, where multi-particle correlation functions (calculations involving the full Kubo formalism) are computationally intensive, or one must rely on single-particle quantities (i.e., Boltzmann formalism), which can be conceptually problematic for strong interactions.

In principle, the high-temperature behavior of the thermopower (thermoelectric power, or Seebeck coefficient) $S$ offers the possibility to directly test the Hubbard model against experiments in strongly correlated materials like the cuprates. Above the Debye temperature, phonons are essentially elastic scatterers of electrons and one might expect thermal relaxation to come overwhelmingly from inelastic scattering off of other electrons. Moreover, room-temperature measurements afford direct contact with determinant quantum Monte Carlo (DQMC)[7,8] simulations, which are limited by the fermion sign problem to temperatures above roughly $J/2$ (half of the spin-exchange energy). Thus, one can address directly an essential question—can the Hubbard model give both qualitative and quantitative agreement with the observed thermopower in cuprates at high temperatures?

Systematic studies of the room-temperature thermopower across a wide variety of cuprates[9–17] show that the thermopower falls roughly on a universal curve over a broad range of hole doping $p$, with a more-or-less universal sign change near optimal doping. This sign change has been interpreted as evidence for a Lifshitz transition[18–20]; however, this implies that the doping associated with the sign change depends

[1]Department of Applied Physics, Stanford University, Stanford, CA 94305, USA. [2]Stanford Institute for Materials and Energy Sciences, SLAC National Accelerator Laboratory, 2575 Sand Hill Road, Menlo Park, CA 94025, USA. [3]Department of Physics and Institute of Condensed Matter Theory, University of Illinois at Urbana-Champaign, Urbana, IL 61801, USA. [4]Department of Physics and Astronomy, University of Notre Dame, Notre Dame, IN 46556, USA. [5]Stavropoulos Center for Complex Quantum Matter, University of Notre Dame, Notre Dame, IN 46556, USA. [6]Department of Materials Science and Engineering, Stanford University, Stanford, CA 94305, USA. [7]Geballe Laboratory for Advanced Materials, Stanford University, Stanford, CA 94305, USA. ✉e-mail: wenwang.physics@gmail.com; tpd@stanford.edu

on material specifics and the detailed shapes of Fermi surfaces, which is hard to reconcile with the observed universality. An alternative interpretation of the sign change appeals to the atomic limit[21–27]; however, the atomic limit requires extremely strong interactions and a very high temperature $T$ compared to the bandwidth, neither of which is satisfied in cuprates at room temperature. The thermopower $S$ also has been approximated by the entropy per density, defined through the Kelvin formula $S_{Kelvin} = (\partial s/\partial n)_T/e^{*}$[28], where charge $e^{*} = -e$ for electrons. $S_{Kelvin}$ is believed to be an accurate proxy for the thermopower $S$, since it accounts for the full effects of interactions, while bypassing the difficulties in exactly calculating the Kubo formula[25,28–30]. However, a direct comparison between $S$ and $S_{Kelvin}$ is required before drawing any conclusions based on these assumptions.

Here, we calculate the thermopower $S$ based on the many-body Kubo formula, as well as the Kelvin formula $S_{Kelvin}$, for the $t$-$t'$-$U$ Hubbard model. We employ numerically exact DQMC and maximum entropy analytic continuation (MaxEnt)[31,32] to obtain the DC transport coefficients that specifically enter the evaluation of $S$. Our results show that the Hubbard model can quantitatively capture the magnitudes and the general patterns of $S$ that have been observed in cuprate experiments.

## Results

The doping dependence of thermopower $S$ from the Hubbard model is shown in Fig. 1 for three different sets of parameters at their lowest achievable temperatures, overlaid with experimental data from several families of cuprates. It is important to note that in the process of converting our results to real units based on universal physical quantities $k_B$ and $e$, there are no adjustable parameters: $S$ is a ratio, so the standard units of $t$ (or $U$) in the Hubbard model factor out. The most striking observation is the surprisingly good agreement between our results and the room-temperature thermopower in cuprates, in both qualitative trend and quantitative magnitudes. Both the simulation and experimental data show a sign change roughly at $p \sim 0.15$. In both cases, $S$—a quantity proportional to the electronic resistivity—increases dramatically in the low doping regime, as the system approaches a Mott

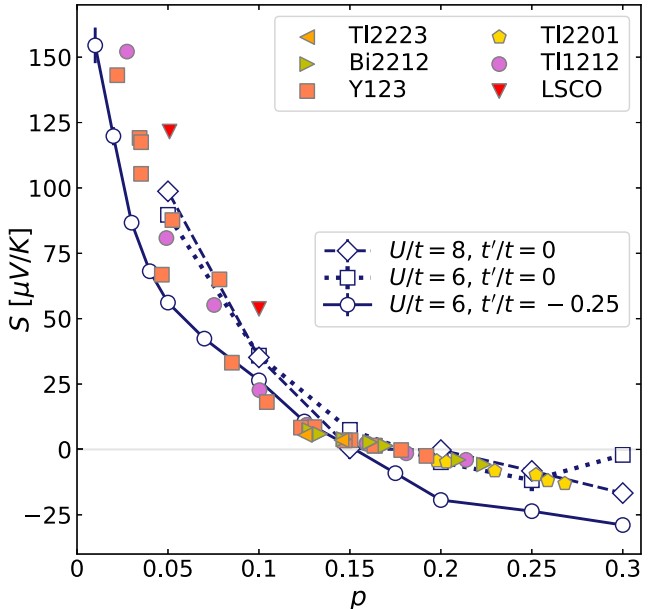

**Fig. 1 | Comparison of simulated and experimental thermopower.** Thermopower $S$ as a function of doping $p$ from DQMC simulations (empty markers connected by lines), compared with doping dependence of $S$ for various cuprates at $T = 290\,K$ (solid scattered markers, data from refs. 9,11). For $U/t = 8$ and $t'/t = 0$, the temperature is $k_B T = t/3.5$. For $U/t = 6$, the temperature is $k_B T = t/4$ for both $t'/t = 0$ and $t'/t = -0.25$.

insulator. The simulation shows moderate $U$ and $t'$ dependence, without significantly affecting agreement with experiments. The moderate parameter dependence is consistent with the observed approximate universality of the doping dependence of the room-temperature $S$ for different cuprates, which may have varying effective $U$ and $t'$.

For weakly interacting electrons, $S$ is expected to change sign around the Lifshitz transition. The sign change in our model with strong interactions, which occurs at $p \sim 0.15$ for $t'/t = -0.25$, is much lower than the Lifshitz transition, which occurs at $p \sim 0.26$ for the same parameters, nor is it associated with the atomic limit (see Supplementary Note 3 and Supplementary Note 5 for details). Therefore, we seek a deeper understanding from $S_{Kelvin} = -(\partial s/\partial n)_T/e$, entropy variation per density variation at a fixed temperature, or equivalently, by the Maxwell relation, $(\partial \mu/\partial T)_n/e$, chemical potential variation per temperature variation at fixed density (see Supplementary Note 4). In Fig. 2, we compare the doping dependence of $S$ and $S_{Kelvin}$. Despite differences in exact values, the sign change of $S$, as shown in Fig. 2a, is closely associated with that of $S_{Kelvin}$, as shown in Fig. 2b. The sign change of $S_{Kelvin}$ occurs when the temperature dependence of the chemical potential $\mu$ vanishes at fixed density—an "isosbestic" point, as exemplified in the inset of Fig. 2b, and highlighted by the arrows.

The doping dependence of $S$ and $S_{Kelvin}$ are also qualitatively similar, and $U$ generally affects both $S$ and $S_{Kelvin}$ in a similar manner, moderately reducing the doping at which each changes sign as $U$ increases. However, $t'$ has more significant and opposite effects on $S$ and $S_{Kelvin}$. Comparing Fig. 2a and b shows us that even though $S_{Kelvin}$, a thermodynamic quantity, differs from $S$, since it does not reflect the dynamics captured by transport[33], $S_{Kelvin}$ still reflects the most important effects from the Hubbard interaction, showing a doping dependence and sign change similar to $S$.

We now examine the temperature dependence of $S$ and $S_{Kelvin}$, using $U/t = 6$ and $t'/t = -0.25$, shown in Fig. 3, as a representative example. The temperature dependence of $S$ in Fig. 3a and $S_{Kelvin}$ in Fig. 3b are qualitatively similar. As temperature decreases from high temperatures, $S$ and $S_{Kelvin}$ first increase, following the atomic limit ($t, t' \ll k_B T, U$, see Supplementary Note 5). As temperature decreases further and passes the scale $t/k_B$, their behaviors deviate from the atomic limit. At low doping ($p \lesssim 0.07$), $S$ and $S_{Kelvin}$ monotonically increase, but at higher doping levels, they first decrease before increasing again down to the lowest temperature, with a dip appearing in between.

We find the dip and the low-temperature increase in both $S$ and $S_{Kelvin}$ particularly interesting, since this upturn commonly appears in cuprates[9–11,13,16], and cannot be understood in either the atomic or weakly interacting limits. To understand its origin, we consider the relationship between $S_{Kelvin}$ and the specific heat $c_v$ using the Maxwell relation $-e(\partial S_{Kelvin}/\partial T)_p = -(\partial c_v/\partial p)_T/T$, where, by definition, $S_{Kelvin} = (\partial s/\partial p)_T/e$ and $c_v = T(\partial s/\partial T)_p$. Specific heat $c_v$ results, also for $U/t = 6$ and $t'/t = -0.25$, are shown in the inset of Fig. 3b. Near half-filling and for temperatures below the spin-exchange energy $J$ ($= 4t^2/U$ to leading order), $c_v$ starts to increase with decreasing temperatures, which is believed to be associated with spin fluctuations[34–36], and $c_v$ drops with increasing doping. Correspondingly, $S_{Kelvin}$ at fixed doping increases with decreasing temperatures, leading to a low-temperature upturn. As the upturn is a common feature shared by $S$ and $S_{Kelvin}$, it is reasonable to believe that the origin should be the same.

The low-temperature slope of the thermopower can be compared with experiments. The negative slopes quoted in ref. 11 for $Bi_2Sr_2CaCu_2O_{8+\delta}$ and $Tl_2Ba_2CuO_{6+\delta}$ range roughly from $-0.05$ to $-0.02\,\mu V/K^2$. Assuming $t/k_B \sim 4000\,K$, this range corresponds to $[-2.3, -0.9]\,k_B^2/(te)$ in our model. We estimate the slope in our model by taking the finite difference between temperatures $k_B T = t/4$ and $t/3.5$ in Fig. 3a and b. For doping between $p = 0.1$ and $0.2$, the calculated slope ranges between $[-2.1, -1.5]\,k_B^2/(te)$ for $S$, and $[-1.8, -0.2]\,k_B^2/(te)$ for $S_{Kelvin}$. Even though systematic and statistical errors in $S$ introduce

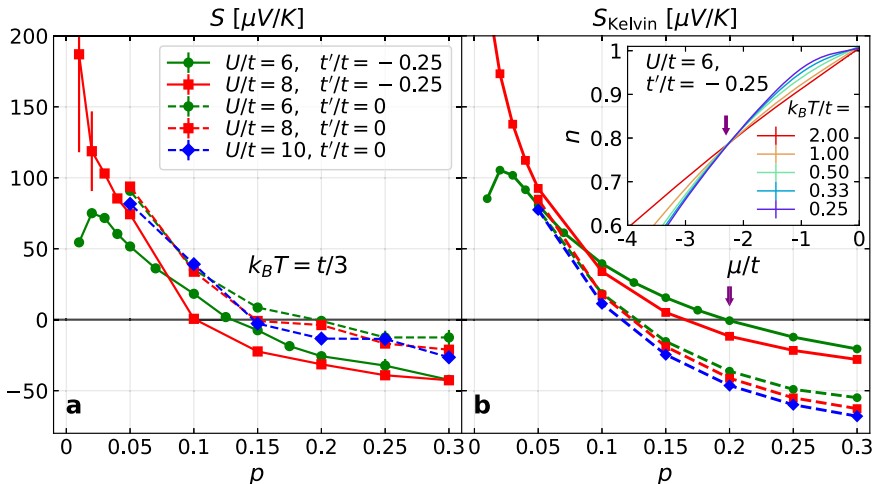

**Fig. 2 | Doping dependence and sign change of $S$ and $S_{\text{Kelvin}}$.** Thermopower $S$ (**a**) and the Kelvin formula for the thermopower $S_{\text{Kelvin}}$ (**b**) as a function of doping $p$ for the Hubbard model with different $U$ and $t'$, all at the same temperature $k_B T = t/3$. Inset of (**b**): density $n$, measured using DQMC, as a function of the chemical potential $\mu$ for $U/t = 6$, and $t'/t = -0.25$ at different temperatures $T$. The arrows in (**b**) and its inset indicate the correspondence between the sign change of $S_{\text{Kelvin}}$ and the vanishing of the temperature dependence of $\mu$ at fixed density.

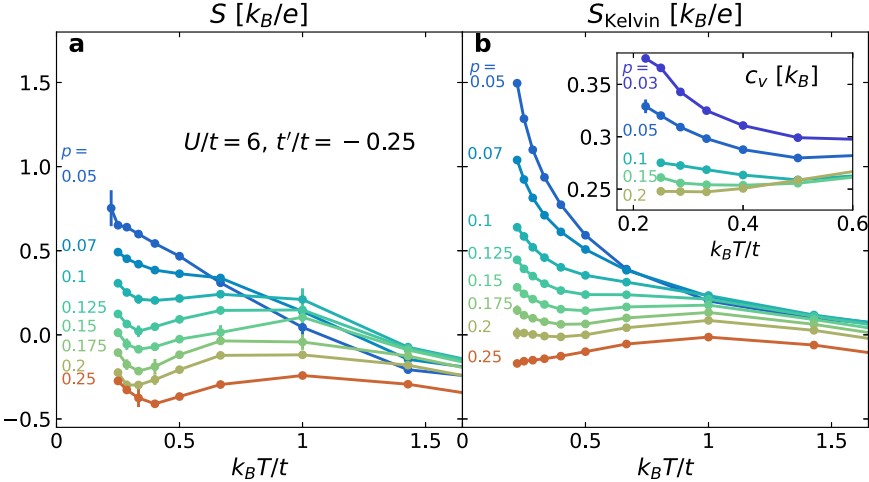

**Fig. 3 | Temperature dependence of $S$ and $S_{\text{Kelvin}}$.** Thermopower $S$ (**a**), and the Kelvin formula for the thermopower $S_{\text{Kelvin}}$ (**b**), as a function of temperature $T$, at different doping levels $p$, for $U/t = 6$, and $t'/t = -0.25$. Inset of **b** shows the specific heat $c_v$ measured using DQMC as a function of temperature for different doping levels.

uncertainties to this slope estimate, the ranges are roughly comparable between simulated $S$, $S_{\text{Kelvin}}$, and experimental values.

For a detailed verification and analysis of the relationship between $S_{\text{Kelvin}}$ and $c_v$, we calculate $-\partial^2 s/(\partial p \partial T)$ from derivatives of independently measured $S_{\text{Kelvin}}$ and $c_v$, for both $U/t = 6$ and $U/t = 8$ with $t'/t = -0.25$, as shown in Fig. 4. Results from the two methods are consistent, up to minor discrepancies such as taking derivatives from discrete data points. At any point along the contour $\partial^2 s/(\partial p \partial T) = 0$ (black solid lines), either a peak or a dip will occur in $S_{\text{Kelvin}}$ as a function of $T$. We observe that a peak appears at temperatures above $J/k_B$ (dashed horizontal line) and a dip appears at temperatures below $J/k_B$. Note that $T \sim J/k_B$ corresponds roughly to the crossover between a peak or dip in $S_{\text{Kelvin}}$ for both $U/t = 6$ and $U/t = 8$ (c.f. Supplementary Fig. 6), supporting our idea that the non-monotonic temperature dependence of both $S_{\text{Kelvin}}$ and $S$ should be associated with effects of spin exchange.

## Discussion

In summary, we calculated the thermopower $S$ and the Kelvin formula $S_{\text{Kelvin}}$ in the Hubbard model. $S$ shows qualitative and quantitative

agreement with the universal curve of the room-temperature $S$ in cuprates, with a sign change corresponding to an "isosbestic" point in $n$ versus $\mu$. $S$ and $S_{\text{Kelvin}}$ show qualitatively similar doping dependence, and the doping at which $S$ changes sign corresponds well to that of $S_{\text{Kelvin}}$. As a function of temperature, we observe a low-temperature upturn in $S$ and $S_{\text{Kelvin}}$ with a slope quantitatively comparable with the corresponding linear increase in cuprates, and we provide evidence supporting their association with the scale of $J$. With this general agreement, we demonstrate that major features in the universal behavior of $S$ in cuprates can be replicated through a quantitative assessment of $S$ in the Hubbard model. The observation that $S_{\text{Kelvin}}$ captures qualitative features of $S$ enables us to understand the experimental thermopower results from the perspective of entropy variation with density.

We emphasize the significance of such a high level of agreement between simulations and experiments for thermopower. Transport properties can be sensitive to numerous factors, which may be different between cuprates and the $t$-$t'$-$U$ Hubbard model. The combination of the model's simple form and capability to reproduce

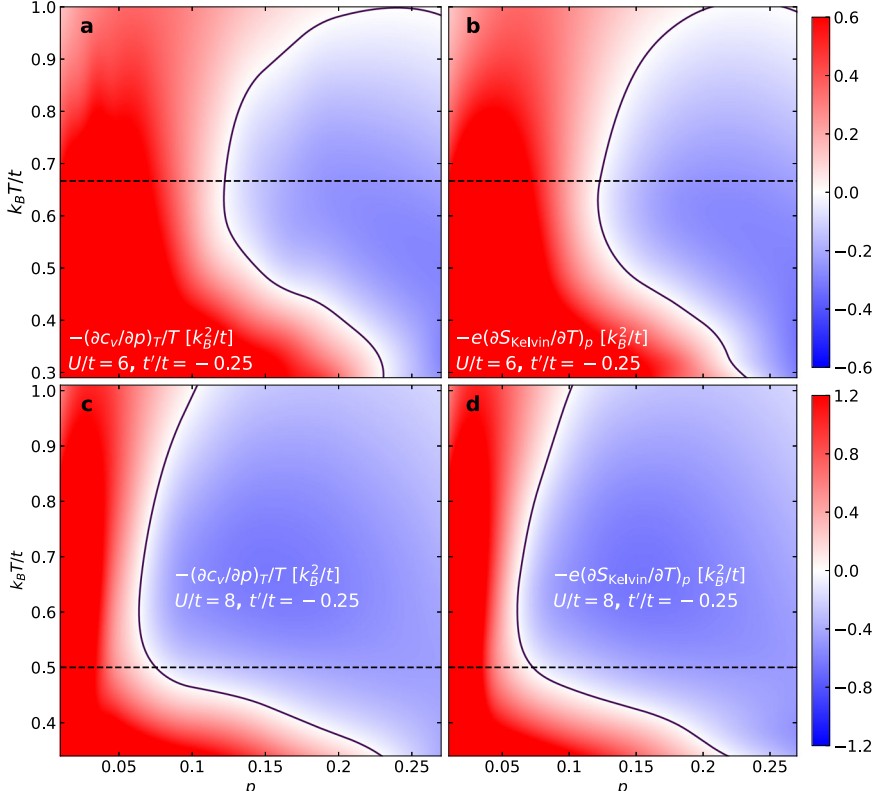

**Fig. 4 | Analysis using $c_v$ and $S_{\text{Kelvin}}$.** Color density plots of $-\partial^2 s/(\partial p \partial T)$ calculated from doping derivative of specific heat [$-(\partial c_v/\partial p)_T/T$, (**a**, **c**)] and temperature derivative of $S_{\text{Kelvin}}$ [$-e(\partial S_{\text{Kelvin}}/\partial T)_p$, (**b**, **d**)], for interaction strengths $U/t = 6$ (**a**, **b**) and $U/t = 8$ (**c**, **d**), both with $t'/t = -0.25$. A cubic-spline fit was applied to curves of $c_v$ versus $p$ and $S_{\text{Kelvin}}$ versus $T$, with corresponding derivatives obtained from the fits. The derivatives $-\partial^2 s/(\partial p \partial T)$ were interpolated (cubic) onto the two-dimensional $(p, T)$ plane. Horizontal dashed lines mark the leading-order approximation for the spin-exchange energy $J = 4t^2/U$, and solid lines mark the contour where $-\partial^2 s/(\partial p \partial T) = 0$.

universal features suggests the dominance of interaction effects in the origin of the systematic behavior in the cuprates. Our observations highlight the importance of pursuing high-accuracy simulations accounting for the full effect of interactions in making progress at understanding these enigmatic materials.

## Methods

We investigate the two-dimensional single-band $t$-$t'$-$U$ Hubbard model with spin $S = 1/2$ on a square lattice using DQMC[7,8]. The Hamiltonian is

$$
\begin{aligned}
H = &-t \sum_{\langle lm \rangle,\sigma} \left( c_{l,\sigma}^\dagger c_{m,\sigma} + \text{h.c.} \right) \\
&-t' \sum_{\langle\langle lm \rangle\rangle,\sigma} \left( c_{l,\sigma}^\dagger c_{m,\sigma} + \text{h.c.} \right) \\
&+ U \sum_l \left( n_{l,\uparrow} - \frac{1}{2} \right) \left( n_{l,\downarrow} - \frac{1}{2} \right),
\end{aligned}
\tag{1}
$$

where $t$ ($t'$) is the nearest-neighbor (next-nearest-neighbor) hopping, $U$ is the on-site Coulomb interaction, $c_{l,\sigma}^\dagger (c_{l,\sigma})$ is the creation (annihilation) operator for an electron at site $l$ with spin $\sigma$, and $n_{l,\sigma} \equiv c_{l,\sigma}^\dagger c_{l,\sigma}$ is the number operator at site $l$ with spin $\sigma$.

The Kelvin formula for the thermopower $S_{\text{Kelvin}}$ can be calculated using DQMC through

$$
S_{\text{Kelvin}} = -\frac{\langle (H - \mu N)N \rangle - \langle H - \mu N \rangle \langle N \rangle}{eT(\langle NN \rangle - \langle N \rangle \langle N \rangle)},
\tag{2}
$$

where $N = \sum_l (n_{l,\uparrow} + n_{l,\downarrow})$ is the total electron number operator, and $\mu$ is the chemical potential.

From the Hamiltonian in Eq. (1), the particle current $\mathbf{J}$ and the energy current $\mathbf{J}_E$ are obtained as[37,38]

$$
\begin{aligned}
\mathbf{J} = &\frac{t}{2} \sum_{l,\boldsymbol{\delta}\in\text{NN},\sigma} \boldsymbol{\delta} \left( i c_{l+\delta,\sigma}^\dagger c_{l,\sigma} + \text{h.c.} \right) \\
&+ \frac{t'}{2} \sum_{l,\boldsymbol{\delta'}\in\text{NNN},\sigma} \boldsymbol{\delta'} \left( i c_{l+\delta',\sigma}^\dagger c_{l,\sigma} + \text{h.c.} \right)
\end{aligned}
\tag{3}
$$

and

$$
\begin{aligned}
\mathbf{J}_E = &\sum_{\substack{l,\boldsymbol{\delta_1}\in\text{NN},\\ \boldsymbol{\delta_2}\in\text{NN},\sigma}} \left( -\frac{\boldsymbol{\delta_1}+\boldsymbol{\delta_2}}{4} \right) t^2 \left( i c_{l+\delta_1+\delta_2,\sigma}^\dagger c_{l,\sigma} + \text{h.c.} \right) \\
&+ \sum_{\substack{l,\boldsymbol{\delta}\in\text{NN},\\ \boldsymbol{\delta'}\in\text{NNN},\sigma}} \left( -\frac{\boldsymbol{\delta}+\boldsymbol{\delta'}}{2} \right) tt' \left( i c_{l+\delta+\delta',\sigma}^\dagger c_{l,\sigma} + \text{h.c.} \right) \\
&+ \sum_{\substack{l,\boldsymbol{\delta_1'}\in\text{NNN},\\ \boldsymbol{\delta_2'}\in\text{NNN},\sigma}} \left( -\frac{\boldsymbol{\delta_1'}+\boldsymbol{\delta_2'}}{4} \right) t'^2 \left( i c_{l+\delta_1'+\delta_2',\sigma}^\dagger c_{l,\sigma} + \text{h.c.} \right) \\
&+ \frac{Ut}{4} \sum_{l,\boldsymbol{\delta}\in\text{NN},\sigma} \boldsymbol{\delta} \left( n_{l+\delta,-\sigma} + n_{l,-\sigma} \right) \left( i c_{l+\delta,\sigma}^\dagger c_{l,\sigma} + \text{h.c.} \right) \\
&+ \frac{Ut'}{4} \sum_{\substack{l,\sigma,\\ \boldsymbol{\delta'}\in\text{NNN}}} \boldsymbol{\delta'} \left( n_{l+\delta',-\sigma} + n_{l,-\sigma} \right) \left( i c_{l+\delta',\sigma}^\dagger c_{l,\sigma} + \text{h.c.} \right) \\
&- \frac{Ut}{4} \sum_{l,\boldsymbol{\delta}\in\text{NN},\sigma} \boldsymbol{\delta} \left( i c_{l+\delta,\sigma}^\dagger c_{l,\sigma} + \text{h.c.} \right) \\
&- \frac{Ut'}{4} \sum_{l,\boldsymbol{\delta'}\in\text{NNN},\sigma} \boldsymbol{\delta'} \left( i c_{l+\delta',\sigma}^\dagger c_{l,\sigma} + \text{h.c.} \right).
\end{aligned}
\tag{4}
$$

To make the notations above clear, NN (NNN) denotes the set of nearest-neighbor (next-nearest-neighbor) position displacements.

Specifically, on the two-dimensional square lattice, NN = $\{+\mathbf{x}, -\mathbf{x}, +\mathbf{y}, -\mathbf{y}\}$ and NNN = $\{+\mathbf{x}+\mathbf{y}, -\mathbf{x}+\mathbf{y}, +\mathbf{x}-\mathbf{y}, -\mathbf{x}-\mathbf{y}\}$, where the lattice constant is set to 1 and $\mathbf{x}$ and $\mathbf{y}$ are unit vectors. Here, if $l$ is an arbitrary site label associated with the position vector $x_l\mathbf{x} + y_l\mathbf{y}$, and $\mathbf{v}$ is a vector adding up arbitrary elements in NN and NNN, the notation $l + \mathbf{v}$ represents a unique site label associated with the position $x_l\mathbf{x} + y_l\mathbf{y} + \mathbf{v}$. The heat current is $\mathbf{J}_Q = \mathbf{J}_E - \mu\mathbf{J}$.

We calculate the thermopower

$$S = -\frac{L_{J_Q,J_x}}{eTL_{J_xJ_x}} \quad (5)$$

using DQMC and MaxEnt[31,32]. Here, $J_{Q,x}$ and $J_x$ are the $x$-components of the heat current operator $\mathbf{J}_Q$ and particle current operator $\mathbf{J}$, respectively. For arbitrary Hermitian operators $O_1$ and $O_2$, the DC transport coefficient $L_{O_1O_2} \equiv L_{O_1O_2}(\omega)\big|_{\omega=0}$, where $L_{O_1O_2}(\omega)$ is determined using the Kubo formula

$$L_{O_1O_2}(\omega) = \frac{1}{N_x N_y \beta} \int_0^\infty dt\, e^{i(\omega+i0^+)t} \int_0^\beta d\tau \langle O_1(t-i\tau)O_2(0)\rangle, \quad (6)$$

where $t$ is real time, without confusion with the hopping matrix elements in the Hamiltonian. Here, $N_x$, $N_y$ are the sizes of the lattice along the $x$ and $y$ directions, respectively, $\beta \equiv (k_B T)^{-1}$, and

$$O_1(t-i\tau) = e^{i(H-\mu N)(t-i\tau)} O_1 e^{-i(H-\mu N)(t-i\tau)}. \quad (7)$$

Detailed derivations for Eqs. (5) and (2) are in Supplementary Note 2 and Supplementary Note 4, respectively. For our calculation, the units for both $S$ and $S_{\text{Kelvin}}$ are $k_B/e \approx 86.17\ \mu V/K$.

## Data availability

The data needed to reproduce the figures can be found at https://doi.org/10.5281/zenodo.8286640.

## Code availability

The source code and analysis routines can be found at https://doi.org/10.5281/zenodo.8286636.

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

## Acknowledgements

We acknowledge helpful discussions with D. Belitz, R. L. Greene, S. A. Kivelson, S. Raghu, B. S. Shastry, R. Scalettar, and J. Zaanen. This work at Stanford and SLAC (W.O.W., J.K.D., B.M., T.P.D.) was supported by the U.S. Department of Energy (DOE), Office of Basic Energy Sciences, Division of Materials Sciences and Engineering. E.W.H. was supported by the Gordon and Betty Moore Foundation EPiQS Initiative through the grants GBMF 4305 and GBMF 8691. Computational work was performed on the Sherlock cluster at Stanford University and on resources of the National Energy Research Scientific Computing Center, supported by the U.S. DOE, Office of Science, under Contract no. DE-AC02-05CH11231.

## Author contributions

W.O.W. conceived the study, performed numerical simulations, conducted data analysis, interpreted the data, and wrote the manuscript. J.K.D., E.W.H., B.M., and T.P.D. assisted in data interpretation and contributed to writing the manuscript.

## Competing interests

The authors declare no competing interests.
