## [Peer Review File · Nature Communications]

Quantitative assessment of the universal thermopower in the Hubbard modelREVIEWER COMMENTS

Reviewer #1 (Remarks to the Author):

The authors study the t - t' - U Hubbard model in hopes of shedding light on some of the mysterious physics of the cuprates. In particular, they study the thermopower in the moderate to high temperature regime as a function of doping. This is done primarily because, since the t - t' - U Hubbard model is such a strongly correlated system, numerical methods are required and, in the temperature regime studied Monte Carlo methods can be reliably utilized.

The Kubo formula is the textbook method for calculating the thermopower at all temperatures but is notoriously difficult and unwieldy. The so-called Kelvin formula for the thermopower, on the other hand, is much easier to consider and might be accurate in the temperature regime considered here. Thus, the authors undertake an important study with a few laudable goals: compare the Kubo thermopower with the Kelvin formula, and to hopefully shed light on the cuprates.

It is found that the two thermopower formulas compare favorably with each other in the temperature regimes studied. Perhaps more importantly, they find both formulas produce results that compare favorably with the cuprates. Since the Kelvin formula is more physically transparent, the authors are able to make some qualitative (as well as quantitative) understanding of the mysterious physics of the cuprates.

The paper is well written, gives adequate context for the work, has an extensive supplemental section to guide further work, and represents significant work and steps forward. The results provide good reason for other workers to consider the Hubbard model and the thermopower, and the Kelvin formula, in studies of the cuprates, and other strongly correlated systems, since it can be provide valuable physical insight in the complicated physics of these systems.

Reviewer #2 (Remarks to the Author):

* Main results:

In this paper, the solve the Hubbard model using the finite-temperature determinant quantum Monte Carlo algorithm, the main measurements are the thermopower S and the

Kelvin formula S_{Kelvin} . Similar behavior of S and S_{Kelvin} are observed, including: (1) the similar doping dependence, the sign change of S occurs at $p \sim 0.15$ for $t'/t = -0.25$ and strong interactions, and it is argued that the value of U and t' affect the sign change point; (2) The comparison of S with experimental data from several families of cuprates is performed; (3) the temperature dependence of S and S_{Kelvin} are examined, the behavior of dip and the low-temperature increase in S is observed, and is argued to be interesting due to similarity of the upturn commonly appears in cuprates; (4) a further derivative of S_{Kelvin} and c_{v} suggest that the non-monotonic temperature dependence of both S_{Kelvin} and S should be associated with effects of spin exchange.

* Significance:

A high level of agreement between simulations and experiments for thermopower S is presented, and the effect of the sensitive factors are studied.

* Flaws in interpretation/conclusion:

These numerical simulations are of course interesting, and their direct comparison to the experimental data is impressive. However, I think there are still some points which are not clearly stated in this paper.

(1) The authors compare their numerical data of S with the experimental ones. However, their simulations are based on different parameters of the Hubbard model, such as shown in Fig.1 for $U/t=6$ and 8 , $t'/t=0$ and -0.25 . It is clear that the choice of U/t and t'/t affects the shape of the curves. So how can the authors make the decision that the numerical data agree with the experiments? If one chooses other values of U/t and t'/t , the numerical curves can go further close or apart from the experimental ones.

(2) Apart from the point (1), the analysis in this paper is not thorough, what is the effect of t' on S ? The authors only study the case $t'=-0.25$. The authors claim that the lowest accessible temperature is $T/t=0.25$, what is the average sign? The author may show some data of the average sign, to let the reader know how deep is the influence of the sign problem. In additional, the authors show the finite size effect by present data on 8×8 and 12×12 lattice, which looks far from enough. Since the temperature is relatively high, the larger lattice size should be affordable, at least for 16×16 . The finite size analysis should be more convincing then.

* Conclusion:

Although the direct comparison of the numerical calculations based on the Hubbard model with the experimental data looks attractive, I think the analysis of the paper is not enough.

Reviewer 1

Reviewer's comment:

The authors study the t - t' - U Hubbard model in hopes of shedding light on some of the mysterious physics of the cuprates. In particular, they study the thermopower in the moderate to high temperature regime as a function of doping. This is done primarily because, since the t - t' - U Hubbard model is such a strongly correlated system, numerical methods are required and, in the temperature regime studied Monte Carlo methods can be reliably utilized.

The Kubo formula is the textbook method for calculating the thermopower at all temperatures but is notoriously difficult and unwieldy. The so-called Kelvin formula for the thermopower, on the other hand, is much easier to consider and might be accurate in the temperature regime considered here. Thus, the authors undertake an important study with a few laudable goals: compare the Kubo thermopower with the Kelvin formula, and to hopefully shed light on the cuprates.

It is found that the two thermopower formulas compare favorably with each other in the temperature regimes studied. Perhaps more importantly, they find both formulas produce results that compare favorably with the cuprates. Since the Kelvin formula is more physically transparent, the authors are able to make some qualitative (as well as quantitative) understanding of the mysterious physics of the cuprates.

The paper is well written, gives adequate context for the work, has an extensive supplemental section to guide further work, and represents significant work and steps forward. The results provide good reason for other workers to consider the Hubbard model and the thermopower, and the Kelvin formula, in studies of the cuprates, and other strongly correlated systems, since it can be provide valuable physical insight in the complicated physics of these systems.

Response:

We thank the reviewer for their comprehensive review, insightful understanding, and positive assessment of our paper. We are encouraged that the reviewer agrees with our viewpoints, indicating that our work aligns well with the broader scientific community's perspectives. We deeply appreciate the positive feedback.

Reviewer 2

Reviewer's comment:

* Main results: In this paper, the solve the Hubbard model using the finite-temperature determinant quantum Monte Carlo algorithm, the main measurements are the thermopower S and the Kelvin formula S_{Kelvin} . Similar behavior of S and S_{Kelvin} are observed, including: (1) the similar doping dependence, the sign change of S occurs at $p \sim 0.15$ for $t'/t = -0.25$ and strong interactions, and it is argued that the value of U and t' affect the sign change point; (2) The comparison of S with experimental data from several families of cuprates is performed; (3) the temperature dependence of S and S_{Kelvin} are examined, the behavior of

dip and the low-temperature increase in S is observed, and is argued to be interesting due to similarity of the upturn commonly appears in cuprates; (4) a further derivative of S_{Kelvin} and c_v suggest that the non-monotonic temperature dependence of both S_{Kelvin} and S should be associated with effects of spin exchange.

* Significance:

A high level of agreement between simulations and experiments for thermopower S is presented, and the effect of the sensitive factors are studied.

* Flaws in interpretation/conclusion:

These numerical simulations are of course interesting, and their direct comparison to the experimental data is impressive. However, I think there are still some points which are not clearly stated in this paper.

Response:

We thank the reviewer for their careful reading and for outlining a clear and concise summary of our work's main results and significance.

To ensure clarity and address the reviewer's concerns, we provide a point-to-point response to each question or comment. We are confident that our paper presents its findings in a clear and rigorous manner, making a valuable contribution to the field and suitable for publication in Nature Communications.

Reviewer's comment:

(1) The authors compare their numerical data of S with the experimental ones. However, their simulations are based on different parameters of the Hubbard model, such as shown in Fig.1 for $U/t = 6$ and 8 , $t'/t=0$ and -0.25 . It is clear that the choice of U/t and t'/t affects the shape of the curves. So how can the authors make the decision that the numerical data agree with the experiments? If one chooses other values of U/t and t'/t , the numerical curves can go further close or apart from the experimental ones.

(2) Apart from the point (1), the analysis in this paper is not thorough, what is the effect of t' on S ? The authors only study the case $t' = -0.25$.

Response:

We thank the reviewer for noticing the dependence of our results on the values of U and t' . In the manuscript, we presented results for representative values of $U/t = 6, 8, 10$ and $t'/t = 0, -0.25$, which are believed to be in the relevant regime of the cuprates.

We understand the reviewers concern about the choice of Hamiltonian values used in our simulations, and whether they can be taken as representative of those to mimic the cuprate phase diagram. Indeed, our choices were bound by restrictions specifically aimed to satisfy this requirement. Given that our starting point is an antiferromagnetic Mott insulator at half filling, a U significantly below 6 should not be chosen, which establishes the lower bound. Selecting a U considerably above 10 would hinder our simulations from reaching adequately low temperatures below the spin-exchange energy J to be comparable with experimental temperature scale.

Figure R1: Thermopower for other values of $t'/t = -0.1, -0.2, -0.3$ along with the original data for $t'/t = 0$ and -0.25 used in Fig. 1 in the manuscript. Parameters: $U/t = 6$, temperature $T/t = 0.25$. Simulation lattice size is 8×8 .

Regarding the choice of t' , we selected two representative values of t' based on the consideration that the Fermi surface topology mimics that of cuprates. We have explored additional t' , appended in Fig. R1; however, as this yields no new insights, we choose not to incorporate it into the manuscript.

Our calculations show weak parameter dependence within the relevant parameter regime, comparable with the material-based variability seen in experiments. Therefore we conclude that, without the need for fine-tuning of parameters, thermopower for the Hubbard model quantitatively agrees to a high degree with experimental data.

Reviewer's comment:

The authors claim that the lowest accessible temperature is $T/t = 0.25$, what is the average sign? The author may show some data of the average sign, to let the reader know how deep is the influence of the sign problem.

Response:

We thank the reviewer for pointing out the issue of fermion sign. Indeed, the average sign does affect our ability to go to lower temperatures. This has been well documented.

Just to name a few of them,

- Ref. [R1] (referenced already in the manuscript as [41]) provides a systematic analysis on the fermion sign in its Fig. S7 using the same main parameter set in our manuscript ($U/t = 6, t'/t = -0.25$ on an 8×8 cluster).
- Ref. [R2] offers a comprehensive data set on the geometry dependence of the DQMC

sign problem for different densities, U strengths, temperatures, and spatial lattice sizes.

- Ref. [R3] delves into the fermion sign problem in its Fig. 4 and Fig. S11 for the same model, both with and without t' .

However, we choose not to include this information since these are readily available, and somewhat repetitive and not crucial for this manuscript.

Reviewer’s comment:

In additional, the authors show the finite size effect by present data on 8x8 and 12x12 lattice, which looks far from enough. Since the temperature is relatively high, the larger lattice size should be affordable, at least for 16x16. The finite size analysis should be more convincing then.

Response:

We thank the reviewer for the careful reading and examination of our finite size analysis. The reviewer’s concern likely arose from subtle discrepancies of S_{Kelvin} in Fig. S7 (b), observed between 8×8 and 12×12 for $n = 0.7$ and $n = 0.8$. Following the reviewer’s suggestion, we have added new data into Fig. S7 (b) for the results of S_{Kelvin} from a 16×16 lattice. We have focused on S_{Kelvin} because it is computationally less intensive.

The results are close to our previous results from the 8×8 and 12×12 lattices, reinforcing our confidence that larger lattice sizes would not alter the observed behaviors.

Reviewer’s comment:

* Conclusion: Although the direct comparison of the numerical calculations based on the Hubbard model with the experimental data looks attractive, I think the analysis of the paper is not enough.

Response:

We appreciate the thoughtful feedback of the reviewer and are encouraged by the recognition of the attractiveness in our direct numerical-to-experimental comparison. As also highlighted by Reviewer 1, our study provides important insights into the complicated physics of strongly correlated systems including cuprates.

In light of the additional data and clarifications provided, we hope to have addressed the reviewer’s concerns regarding the completeness of our analysis. Given the significance of our findings and the depth of our analysis, we respectfully request the reviewer’s support for our study’s publication in Nature Communications.

References

[R1] Edwin W. Huang et al. “Strange metallicity in the doped Hubbard model”. In: *Science* 366.6468 (2019), pp. 987–990. DOI: 10.1126/science.aau7063.

- [R2] V. I. Iglovikov, E. Khatami, and R. T. Scalettar. “Geometry dependence of the sign problem in quantum Monte Carlo simulations”. In: *Phys. Rev. B* 92 (4 2015), p. 045110. DOI: 10.1103/PhysRevB.92.045110.
- [R3] R. Mondaini, S. Tarat, and R. T. Scalettar. “Quantum critical points and the sign problem”. In: *Science* 375.6579 (2022), pp. 418–424. DOI: 10.1126/science.abg9299.

List of changes

- In the supplementary material, data for the 16×16 lattice size has been added to Fig. S7 (b). The corresponding information and discussion have been integrated into the figure caption and the “Finite size and Trotter error” section. Error bar information has been incorporated into the “Simulation parameters” section.
- In Fig. S7, horizontal reference lines at $S_{\text{Kelvin}} = 0$ have been added in (a), (b), and (c) to ensure consistency with (d), (e), and (f).
- Acknowledgements to D. Belitz and R. Scalettar have been added.
- In the “Formalism” section of the supplementary material, the phrase “ $\omega = 0$ limit” has been revised to “ $\omega = 0$ value” for greater accuracy.
- A reference (Phys. Rev. B 82, 214503) has been added to both the main text and the supplementary material.
- Sections on code and data availability have been included separately and the links have been updated. The code and data have been prepared and will be published via the provided links by the authors upon reaching the possible acceptance or proof stage.
- Sections detailing author contributions and competing interests have been included.

REVIEWERS' COMMENTS

Reviewer #1 (Remarks to the Author):

I am satisfied with the authors' responses to my report (I had only minor comments) and the other referee's report. The calculations are well motivated, the parameters are well chosen, and the system sizes adequate. It is not really possible to do detailed systematic finite size scaling or analysis with systems. The additional 16x16 cluster results provide more evidence and confidence in the author's conclusions. I recommend publication of the manuscript in the current form.

Reviewer #2 (Remarks to the Author):

The author has satisfactorily answered my questions, additional information are added. I agree the publication in its current form.